# Do Crude Oil Prices Drive the Relationship between Stock Markets of Oil-Importing and Oil-Exporting Countries?

**Manel Youssef** [1] **and Khaled Mokni** [1,2,*]

1   Department of Statistics and Quantitative methods, College of Business Administration,
    Northern Border University, Arar 91431, P.O. Box 1321, Kingdom of Saudi Arabia
2   Department of Quantitative methods, Institut Supérieur de Gestion de Gabès, University of Gabès,
    Street Jilani Habib, Gabès 6002, Tunisia
*   Correspondence: Khaled.Mokni@isgs.rnu.tn or kmokni@gmail.com; Tel.: +966-553-981-566

**Abstract:** The impact that oil market shocks have on stock markets of oil-related economies has several implications for both domestic and foreign investors. Thus, we investigate the role of the oil market in deriving the dynamic linkage between stock markets of oil-exporting and oil-importing countries. We employed a DCC-FIGARCH model to assess the dynamic relationship between these markets over the period between 2000 and 2018. Our findings report the following regularities: First, the oil-stock markets' relationship and that between oil-importing and oil-exporting countries' stock markets themselves is time-varying. Moreover, we note that the response of stock market returns to oil price changes in oil-importing countries changes is more pronounced than for oil-exporting countries during periods of turmoil. Second, the oil-stock dynamic correlations tend to change as a result of the origin of oil prices shocks stemming from the period of global turmoil or changes in the global business cycle. Third, oil prices significantly drive the relationship between oil-importing and oil-exporting countries' stock markets in both high and low oil-stock correlation regimes.

**Keywords:** oil prices; dynamic conditional correlations; oil-exporting countries; oil-importing countries; stock markets

**JEL Classification:** C32; C51; G15; Q40

## 1. Introduction

Over the last two decades an abundant literature, investigating the interconnections between oil and stock markets has emerged. In this context, early influential studies have identified a negative relationship between oil prices and stock market returns (Jones and Kaul 1996; Sadorsky 1999; Nandha and Faff 2008; Miller and Ronald 2009; Chen 2010). On the other hand, several studies show that the responses of stock markets to oil shocks depend on the net position of the country in the global oil market and the driving forces of the oil price shocks. Thus, researchers suggest that positive linkages between oil and stock market returns pertain to oil-exporting countries, while negatives ones are registered for oil-importing countries (Bashar 2006; Mohanty et al. 2011 and Wang et al. 2013).

Furthermore, the oil market represents a profitable alternative destination for many investors and financial institutions regarding the low correlation between oil prices and traditional asset classes and the positive co-movement with inflation, (Kat and Oomen 2007; Silvennoinen and Thorp 2013). In addition, recent studies suggest that oil and stock markets are likely to become highly linked due to the financialization of the oil market, resulting from increased participation and speculation of hedge

funds in this market (Buyuksahin et al. 2010; Silvennoinen and Thorp 2013; Tang and Xiong 2012; Buyuksahin and Robe 2011; Hamilton and Wu 2012; Sadorsky 2014).

Recently, a strand of studies has focused on the dynamic correlation between oil and stock returns (Miller and Ronald 2009; Reboredo 2010; Filis et al. 2011; Daskalaki and Skiadopoulos 2011; Chang and Yu 2013; Reboredo and Rivera-Castro 2014; Zhang and Li 2014; Boldanov et al. 2016; Zhu et al. 2017; Aydogan et al. 2017) given that correlations (covariance) have important implications for asset allocation and portfolio optimization. Overall, despite the increased interest in the oil-stock relationship, the literature has remained relatively silent about the role of oil prices in deriving linkages between stock markets. On other worlds, there is no empirical evidence about the role of oil-stock linkages in predicting the relationship between stock markets in oil-related economies.

Thus the main contribution of this study is to address this issue. In this context, we analyze the role of oil market in driving stock markets relationship in oil-related countries. As shown in Figure 1, first, we investigate the time-varying conditional correlations between oil prices and stock market returns in major oil-exporters and oil-importers. Second, we use these oil-stock dynamic correlations to examine their influential role in determining the linkage between stock market returns in these countries. Then, an accurate understanding of the interrelation between these markets (oil and stock markets) will be valuable for investors and policymakers since oil prices represent an information flow. In recent portfolio theory, diversification strives to reduce the portfolio's unsystematic risk events. However, economic consequences and the risk spillovers arising from declining oil prices could make portfolio diversification more difficult. Therefore, it is often known that risky assets tend to be strongly correlated in the stressed period, which can amplify the risk of collapse (Trabelsi 2017). Thus, to help investors to avoid future losses in their portfolio, it is important to investigate the impact of oil price shocks in stock market returns for oil-exporting and oil-importing economies and identify the role of oil price shocks in driving linkages between these markets.

There is a trend in the financial literature for the time-varying correlations between oil prices and stock markets. The present paper contributes to this trend in many ways. First, we extend the research area dealing with oil-stock linkages in major oil-importing and oil-exporting countries. Given that most previous studies have been conducted across different oil-importing countries, the recent global financial crises and the downward trend of oil prices in recent years have highlighted effects on oil-exporting countries. Previous empirical studies show that some oil-exporting countries (such as Venezuela, Nigeria, and others) share some specific structural economic features; they differ on their reliance on oil price changes (Arouri and Fouquau 2009). Thus, we can conduct a comparison between our results and others relative to most previous studies in this research area. Second, the DCC-FIGARCH[1] framework has benefits compared to the vector autoregressive VAR model[2]. Its use in this study can help us to deeply analyze the dynamic linkages between global oil market and stock market returns. Third, to the best of the authors' knowledge, this is the first paper that investigates the role of oil-stock conditional dynamic correlation, for oil-importing and oil-exporting countries, to predict the dynamic conditional correlations between stock market returns for these countries. Moreover, previous studies focusing on the relationship between oil and stock markets did not provide any idea about the effect of oil price fluctuations in deriving the relationship between stock markets, especially in oil-related economies. This paper is intended to fill the void in the literature. In fact, it is of great importance for investors, decision-makers, and risk managers to understand factors

---

[1]  Empirical studies including: (Youssef et al. 2015); (Mokni and Mansouri 2017), show that the FIGARCH models are able to capture different volatility stylized facts frequently observed in the financial time series such as: volatility clustering heteroscedasticity and long memory at the same time.

[2]  Several empirical studies including, (Stock and Watson 2001) and (Dagher and Hariri 2013), indicate that the VAR methods nonlinearities and conditional heteroscedasticity meanwhile the causality test cannot examine the magnitude of return linkages.

relative to oil price changes that can affect connections between stock markets in order to help guide financial and investment decisions.

The remainder of this paper is organized as follows: Section 2 provides a brief overview of the theory and a summary of previous studies. Section 3 explains the methodology employed. Section 4 presents the data; major empirical findings are introduced in Section 5. Section 6 presents the main policy implications, and Section 7 summarizes and concludes the paper.

## 2. Theory and Literature Background

Economic theory suggests that any asset price should be determined by its expected discounted cash flows (Williams 1938; Fisher 1930). Thus, any factor that should alter these discounted cash flows should have a significant effect on this asset prices (Filis et al. 2011). Consequently, an increase in oil prices would result in a reduction in production, as inputs become more expensive and contribute directly to the level of inflation, which fosters a decrease in investors' earnings expectations from the stock market (Hamilton 1996; Sadorsky 1999; Al-Fayoumi 2009; Arouri and Nguyen 2010). Hence, any rise of oil price should be accompanied with a decline in stock prices. Should this response of stock prices to an oil price increase be similar for both oil-importing and oil-exporting economies?

In the existing literature, many studies claim that oil price shocks influence stock markets indirectly through macroeconomic variables such as inflation and economic growth. (Bjørnland 2009) and (Jimenez-Rodriguez and Sanchez 2005), suggest that a rise in the oil price is expected to have a positive impact in an oil-exporting country, as the country's income will increase. Consequently, the rising income is expected to generate a rise in expenditures and investments, which, in turn, creates greater productivity and unemployment (Filis et al. 2011). In this case, an oil price increase positively affects the stock markets' response.

In contrast, for an oil-importing country, an increase in oil prices is expected to have an opposite effect (see Hooker 1996). In fact, an oil price increase will result in an increase in production costs, since oil is considered as the most important production input (Arouri and Nguyen 2010; Kim and Loungani 1992). The increasing cost will affect consumer's behavior, which will, in turn, decrease their demand and, thus, spending, due to higher consumer prices (Bernanke 2006); (Abel and Bernanke 2001); (Hamilton 1996); (Hamilton 1988a, 1988b); (Barro 1984). Decreasing consumption would result in decreasing production and, in return, increasing unemployment (see, Lardic and Mignon 2006); (Brown and Yücel 2002); (Davis and Haltiwanger 2001). In this case, stock markets are expected to decline (see Sadorsky 1999; Jones and Kaul 1996).

Moreover, we should not ignore the impact of oil price shocks on stock markets due to the uncertainty they create for the financial world, depending on the forces pushing up oil prices (demand-side or supply-side). In fact, stock markets are expected to respond positively to oil price shocks originating from an increase in global demand, and negatively if the shock originates from the supply-side (Filis et al. 2011); (Hamilton 2009b); (Kilian and Park 2009).

Having discussed the possible transmission channels of oil price shocks to stock markets, we move to discuss briefly the anterior literature studies related to the current research area.

A strand of the literature reports that oil price changes have a significant negative impact on stock markets (Jones and Kaul 1996; Ciner 2001, 2013; Papapetrou 2001; Basher and Sadorsky 2006; Driesprong et al. 2008; Chen 2010; Park and Ratti 2008; Miller and Ronald 2009). However, some researchers have confirmed that this adverse relation only applies to oil-importing countries, whereas, positive impacts of oil price movements in stock markets pertains to oil-exporting countries (Sadorsky 2001; El-Sharif et al. 2005; Bashar 2006; Boyer and Filion 2007; O'Neill et al. 2008; Mohanty et al. 2011; Mohanty and Nandha 2011; Arouri and Rault 2012; Filis and Chatziantoniou 2013; Wang et al. 2013). Additionally, some studies including (Lescaroux and Mignon 2008); (Cong et al. 2008); (Apergis and Miller 2009); (Al-Fayoumi 2009); and (Al Janabi et al. 2010) find that oil price changes have little or no effect on stock market returns. Such a result can be explained by the fact that economies are less vulnerable to the influences of oil price changes and, thus, these effects are no longer transmitted to

stock markets. (Filis et al. 2011) indicate that currently, monetary authorities put emphasis on inflation stability and, thus, they prevent any inflationary pressures caused by oil price changes. Consequently, they reduce the effects of oil price changes in the economy and, thus, on the stock market.

Furthermore, the (International Energy Agency 2006); (Nordhaus 2007) and (Blanchard and Gali 2007) suggest that recent developments in investment, production, wage policies, and renewable resources tend to minimize the consequences of oil price changes in economies generally and stock markets particularly. Nevertheless, several studies show that the impact of oil price changes in the national economies of oil-importing-countries can differ from those of oil-exporting countries. While the relationship between oil price changes and microeconomic activities has always been reported as negative, an escalation in oil prices can induce positive effects on the national economies of oil-exporting countries. Conversely, an increase in oil prices can induce increases in industry costs and inflation rates, as well as a reduction in expenditures in non-oil goods (Barsky and Kilian 2004) in oil-importing countries. Oil price increases may generate a substantial income in oil-exporting countries as a result of the low price elasticity of crude oil demand (Bjørnland 2009; Jung and Park 2011).

Given the heterogeneous effects, the response of stock market returns to oil price shocks in oil-exporting countries may be determined by the relative significance of positive and negative impacts in these countries (Wang et al. 2013). In the existing literature, a large number of studies have investigated the linkage between oil price shocks and stock markets in oil-importing countries, and especially the U.S market as the largest oil importer. There is a consensus among these studies about the evidence of the negative relationship between oil and stock market activities (Basher et al. 2012; Chen 2010; Elder and Serletis 2010; Jones and Kaul 1996; Kilian and Park 2009; Masih et al. 2011; Sadorsky 1999; Wei 2003). Nonetheless, few studies show that the response of stock markets to oil price movement is not always as significant as it is generally expected (Huang et al. 1996; Apergis and Miller 2009; Miller and Ronald 2009). On the other hand, the same studies considered oil-exporting countries and focused on studying the linkages between oil price changes and stock market activities in these countries. (Sadorsky 2001) investigate the relationship between oil prices and equity values in the Canadian oil and gas sector. He found a significant positive relationship between oil price changes and the oil equity index in this country. (Park and Ratti 2008) and (Bjørnland 2009) show that an increase in oil prices affects positively the Norwegian stock market as an oil-exporting country. More recently, (Mohanty et al. 2011), found that, except, for Kuwait, oil price changes have a positive impact on the stock returns in GCC countries. In the same context (Demirer et al. 2015) found that, for the Gulf Arab stock markets, higher oil prices lead to higher stock values. (Filis et al. 2011); and (Guesmi and Fattoum 2014), among others, found that stock market responses to oil price changes depend on their countries' net position in the global oil market.

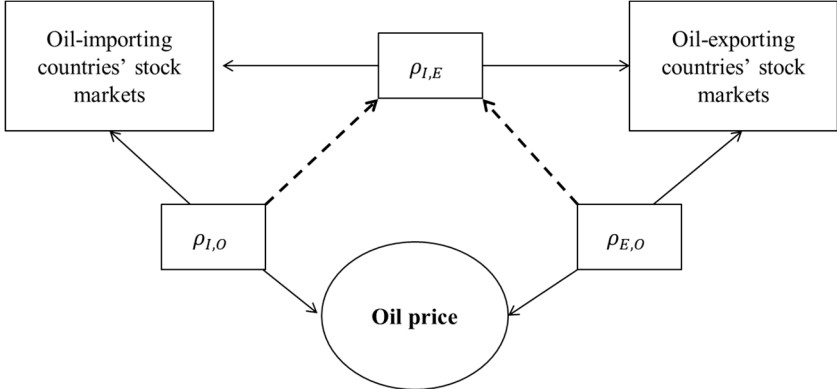

**Figure 1.** Relationship between correlations.

Furthermore, an array of studies found that the responses of stock market returns to oil price changes are asymmetric and time-varying (Miller and Ronald 2009; Reboredo 2010; Filis et al. 2011);

(Daskalaki and Skiadopoulos 2011); (Chang and Yu 2013); (Reboredo and Rivera-Castro 2014); (Zhang and Li 2014). (Choi and Hammoudeh 2010) employed a DCC model to investigate the correlation between stock and oil markets (among other commodities), since the Iraq war in 2003. They report that correlations are negatives and decreasing over time. Similarly, (Bhar and Nikolova 2010) examined the correlation between the Russian stock market and oil market and found evidence of a permanent negative correlation between oil and the Russian stock market during the 1995–2007 period. (Broadstock et al. 2012) employ a BEKK model to examine the time-varying correlation between oil price changes and energy-related equities in China. Evidence from this study suggests that correlation increased significantly during the global financial crisis. (Broadstock and Filis 2014) used a scalar-BEKK model to investigate the dynamic relationship between oil prices and stock markets in the U.S and China. Using the same methodology, (Filis 2014) investigated the co-movements between oil prices and a range of oil-exporting and oil-importing countries. Their findings are similar, indicating that the U.S. stock market has higher correlations with oil prices over time compared to other importing countries. Furthermore, they show that correlations fluctuate between negative and positive values depending on the considered period. In the same context, (Boldanov et al. 2016), employed a diag-BEKK model to investigate the time-varying conditional correlation between oil price and stock market volatility for six oil-exporting and oil-importing countries. They found that the correlation between oil and stock market volatilities changes over time and fluctuates between negative and positive values. Additionally, heterogeneous patterns in the time-varying correlation are evident between oil-exporting and oil-importing countries. (Aydogan et al. 2017), employ a cDCC-GARCH model to investigate the impact of oil price changes on stock markets in oil exporter and oil importer countries. They found that the time-varying correlations between oil and stock prices for oil-importing countries are more pronounced than for oil-exporting countries. This finding confirms anterior results indicating that the correlation between the volatilities of the stock market and oil price returns vary depending on the net position of the country in the global oil market.

## 3. Methodology

In this paper, the methodology consists of two stages. The first is the estimation of the dynamic conditional correlations by using the DCC-FIGARCH model. Second, we employ the estimation of these correlations to test the role of the oil price to determine the linkage between stock markets in oil-importing and oil-exporting countries.

### 3.1. Modeling the Dynamic Conditional Correlations: DCC-FIGARCH Model

In order to estimate the time-varying dynamic conditional correlation between oil prices and stock markets returns and between stock markets themselves, we use the dynamic conditional correlations GARCH (DCC-GARCH) model introduced by (Engle 2002). According to (Engle 2002), this model provides a very good approximation to a variety of time-varying correlation processes. Additionally, the comparison of DCC with simple multivariate GARCH and several other estimators shows that the DCC is often the most accurate (moreover, it requires the estimation of fewer parameters than other multivariate GARCH models (Filis et al. 2011)).

The estimation includes two steps. At the first step, the conditional variances of the assets $\sigma_t^2$ are estimated with a GARCH process. GARCH models, early introduced by (Bollerslev 1986) to generate conditional volatility for daily log-returns, have shown some limitations, including the incapability to capture some stylized facts such as asymmetry and persistence in financial series. To overcome these drawbacks, (Baillie et al. 1996) proposed the fractionally integrated GARCH (FIGARCH) process in order to capture the phenomenon of long memory frequently observed in financial series returns. Since their introduction, the FIGARCH processes have been largely employed in researches to model financial series (see: Youssef et al. 2015; Mokni and Mansouri 2017; Mokni 2018). These studies highlight that the fractionally integrated GARCH processes are more appropriate to filter the financial return series. Following the aforementioned studies, we employ the univariate FIGARCH(1,d,1) to

take into account the different stylized facts, aforesaid, frequently observed in financial return series. Moreover, the innovation series are assumed to follow the Student's *t* distribution, firstly proposed in the estimation of multivariate ARCH proess models by (Harvey et al. 1992) and (Fiorentini et al. 2003). The Student's *t* distribution allows modeling the excess leptokurtosis which is not captured by the ARCH process[3].

The conditional variances of assets issued from this specification are expressed as follows:

$$y_t = u_t + \varepsilon_t; \ \varepsilon_t = z_t \sqrt{\sigma_t^2} \tag{1}$$

$$\sigma_t^2 = \omega + \left[1 - \beta L - (1 - \phi L)(1 - L)^d\right]\varepsilon_t^2 + \beta \sigma_{t-1}^2; \tag{2}$$

$$z_t \sim t(0, 1; v) \tag{3}$$

In this study, we use the dynamic conditional correlation (DCC) model introduced by (Engle 2002). Let $r_t$ be the vector composed of two returns series, $r_t = (r_1, \ r_2)'$. Denoting the lag polynomial $A(L)$, we have:

$$A(L)r_t/\mathcal{F}_{t-1} = \mu_t + \varepsilon_t \tag{4}$$

where $\varepsilon_t$ is the error-term vector The DCC model is based on the hypothesis that the conditional returns are normally distributed with zero mean and conditional covariance matrices $H_t = E[r_t r'|I_{T-1}]$. the covariance matrix is expressed as follows:

$$H_t \equiv D_t R_t D_t \tag{5}$$

where $D_t = \text{diag}\left\{\sqrt{H_{it}}\right\}$ is a diagonal matrix of time-varying standard deviations issued from the estimation of univariate FIGARCH processes described by Equation (2).

$R_t$ is the conditional correlation matrix of the standardized disturbances $\varepsilon_t$, where $\varepsilon_t = D_t^{-1}r_t$:

$$R_t = \begin{bmatrix} 1 & \rho_{12,t} \\ \rho_{12,t} & 1 \end{bmatrix} \tag{6}$$

The matrix $R_t$ is decomposed into:

$$R_t = Q_t^{*-1}Q_tQ_t^{*-1} \tag{7}$$

where $Q_t$ is the positive definite matrix containing the conditional variances-covariances of $\varepsilon_t$ and $Q_t^{*-1}$ t is the inverted diagonal matrix with the square root of the diagonal elements of $Q_t$:

$$Q_t^{*-1} = \begin{bmatrix} 1/\sqrt{q_{11,t}} & 0 \\ 0 & 1/\sqrt{q_{22,t}} \end{bmatrix} \tag{8}$$

The DCC model is then given by:

$$Q_t = (1 - a - b)\overline{Q} + a\epsilon_{t-1}\epsilon'_{t-1} + bQ_{t-1} \tag{9}$$

where $\overline{Q}$ is the unconditional covariance of the standardized disturbances $\varepsilon_t$ The dynamic conditional correlations are finally given by:

$$\rho_{ij,t} = \frac{q_{12,t}}{\sqrt{q_{11,t}q_{22,t}}} \tag{10}$$

---

[3]  The degree of leptokurtosis induced by the ARCH process does not capture all of the leptokurtosis present in the log-returns. Thus, there is strong evidence that the conditional distribution of the innovations series is not-normal. For further details see (Xekalaki and Degiannakis 2010).

Note that, following (Engle 2002), the estimation of this model is done using a two-step maximum likelihood estimation method, in which the likelihood function is given by:

$$\ln(L(\theta)) = -\frac{1}{2} \sum\nolimits_{t=1}^{T} n \ln(2\pi) + \ln|D_t|^2 + \ln(|R_t|) + \varepsilon_t' D_t^{-2} \varepsilon_t \tag{11}$$

### 3.2. The Role of Crude Oil Prices

In this paragraph, we test if the degree of linkage of a stock market to oil price changes could affect its relationship with another stock market. In other words, we investigate the mediator effect of crude oil on determining the relationship between oil-exporting and oil-importing stock markets. Using the dynamic correlations between oil and the stock market of each category of countries, the baseline empirical equation incorporating the potential mediator effect of crude oil is as follows:

$$\rho_t^{E,I} = \alpha_0 + \alpha_1 \rho_t^{E,WTI} + \alpha_2 \rho_t^{I,WTI} + \varepsilon_t \tag{12}$$

where E, I denote oil-exporters and oil-importers, respectively; $\rho_t^{E,I}$ is the dynamic conditional correlation between oil-exporters and oil-importers' stock markets at the time t. $\rho_t^{E,WTI}$ denotes the dynamic conditional correlation between oil and oil-exporters stock markets at time t, and $\rho_t^{I,WTI}$ is the conditional correlation between oil and the oil-importers' stock market at time t. Figure 1 shows the different relationships between correlations investigated in this study.

Policymakers and investors are eager to deeply understand how oil-stocks correlations affect the relationship between stock markets of oil-importing and oil-exporting countries under different market conditions to make more detailed strategies for risk management and investments. Additionally, numerous studies in the dependence modeling literature including, (Longin and Solnik 1995); (Ramchand and Susmel 1998); (Ang and Bekaert 2002); (Reboredo 2012); (Tang and Xiong 2012); (Creti et al. 2013); and (Mokni and Mansouri 2017), suggest that the dependence between financial assets would be more likely to increase during periods of extreme market movements. Therefore, one can argue that the model expressed by the Equation (12) can be affected by different markets conditions. Thus, a significant weakness of the model in Equation (12) is that it is static, i.e., the parameters are assumed to be constant during the whole sample period, ignoring possible structural breaks in conditional correlations. For this purpose, we extend this model to a regime-switching framework in order to differentiate between low and high market correlation states. Formally, we estimate a two-state Markov-Switching (MS) model in the form:

$$\rho_t^{E,I} = \alpha_{0,S_t} + \alpha_{1,S_t} \rho_t^{E,WTI} + \alpha_{2,S_t} \rho_t^{I,WTI} + \varepsilon_{t,S_t} \; ; \; \varepsilon_{t,S_t} \sim N\left(0, \sigma_{\varepsilon,S_t}^2\right) \tag{13}$$

where $S_t \in \{1, 2\}$ follows the first-order two states Markov Switching process. The first state is related to the normal market regime or normal correlations periods, while the second represent the stress market or high correlations regime.

## 4. Data

### 4.1. Selected Countries

As a cross-country study, we select six countries with respect to their position in international oil markets and economies to represent deferent oil dependence levels, according to the volume of oil exports and imports and stock market indicators relative to the crude oil prices. Table 1 presents the selected countries based on their dependence on oil (exporters/importers). Information about crude oil imports and exports are sourced from the "World to exports" (http://www.worldstopexports.com) website and stock markets statistics are sourced from the "World Data Atlas" website (https://knoema.com/atlas) for the year 2017. In this section we briefly review the selected countries to reveal their economic position in the international oil market.

**Table 1.** Countries selected based on their oil market status.

| Country | 2017 Crude Oil Exports/Imports (Billion USD) | % World Total | Stock Market Capitalization (Billion USD) | % GDP |
|---|---|---|---|---|
| *Oil-importing countries* | | | | |
| Russia | 93.3 | 11.10% | 623.4 | 39.5% |
| Canada | 54 | 6.40% | 2367.1 | 143.2% |
| Norway | 25.9 | 3.10% | 287.2 | 72.0% |
| *Oil-exporting countries* | | | | |
| China | 162.2 | 18.60% | 8429.9 | 71.2% |
| United States | 139.1 | 15.90% | 32,120.7 | 165.7% |
| Japan | 63.7 | 7.30% | 6222.9 | 127.7% |

Note: The table presents the major oil-importing and exporting countries, selected in the current study, based on the volume of exportation and importation of each country for the year 2017, the proportion of the importation (exportation) volume of the world total importation (exportation), the stock market capitalization expressed in billion USD and the oil exportation (importation) as a percentage of gross domestic production (GDP). Source: Crude oil imports and exports are sourced from the "World to exports" website (http://www.worldstopexports.com). Stock markets statistics are sourced from the "World Data Atlas" website (https://knoema.com/atlas).

Russia exported $93.3 billion barrel per day (bpd) in 2017, which accounts for 11.10% of the world total exports and for nearly 40% of the gross domestic product (GDP) of the country. Accordingly, the country is highly dependent on hydrocarbons; Russia was the second-largest oil exporter in the world following Saudi Arabia. The stock market index that we focus in this study is the RTS index, which contains the 50 largest companies listed on the Moscow Exchange (the majority of the equities in the index pertain to oil and gas industry).

Canada, among the world's five largest oil producers, ranks fourth in the world in terms of proven oil reserves. The country exports $54 billion (bpd), which accounts for 6.4% of the total world exports and about 144% of the country's GDP. The country is a net exporter of energy commodities, and is the principal source of energy imports for the U.S. In the current study we use the S&P/TSX composite index that represent the largest companies listed on the Toronto stock exchange.

The Norwegian economy is highly dependent on the petroleum sector. The country exported about $26 billion (bpd) in 2017, which accounted for the largest portion of exports revenues, and represents 72% of the country's GDP, making it the eleventh largest oil exporter and the main European producers. To represent the Norwegian stock market we use the OSEAX (OBX) index, which represents all the equities listed on the Oslo stock Exchange (EIA 2015b).

The three oil-importing countries considered in this study are: the U.S, Japan and China. China is the world's largest energy consumer and producer due to its status as the world's most populous country, with a fast growing economy, (Kayalar et al. 2017). It was the largest oil consumer and oil importer in 2017, with total imports of $162.6 billion (bpd), which accounted for 18.6% of total oil-importation. Despite its extensive oil fields, China has faced increased pressure to import greater volumes from different sources, because of substantial oil demand growth and geopolitical uncertainties.

The U.S ranked as the world's second largest oil-importer with volume of oil importation of $139.1 billion (bpd) in 2017, which represented 15.9% of the world's total oil imports.

The Japanese economy is the third largest oil-importer and consumer. The country's net imports of crude oil not counting domestic oil production were $63.7 billion (bpd) in 2017. Nonetheless, Japan experienced dramatic growth in oil imports after the Fukushima incident and earthquakes in the middle of 2011. The loss of nuclear power generating plants led to use crude oil for direct burning in electricity plants (EIA 2015). On the other hand, Japanese government maintains control of oil stocks to guarantee stable crude oil supply in the event of import disruption. Moreover the Japanese economy is highly dependent on the Middle East and, therefore, trying to diversify its supply sources in Russia, Southeast Asia, and West Africa.

In the current study, the oil-importing countries, China, U.S., and Japan are presented by the Shanghai Composite (SSC), the S&P500, and the NIKKEI225, respectively.

### 4.2. Preliminary Analysis and Descriptive Statistics

Our analysis is based on the daily closing spot prices for WTI crude oil, which is a global benchmark for determining the prices of other light crudes in the United States, and six market indices for three oil-exporting (OBX (Norway), TSX (Canada), and RTS (Russia), and three major oil-importing-countries (S&P500 (U.S), NIKKEI (Japan), and SSE (China)). The period of the study spans from 6 January 2000 to 15 October 2018. The data for stock markets indices are sourced from Datastream, while the data for the WTI oil price is obtained from the US Energy Information Administration (EIA). The chosen period permits the investigation of the oil-stock interrelations during extreme economic and geopolitics events such as the early-2000 recession, the 9/11 terrorist attacks, the global financial crisis of 2007–2008 and the 2009–2012 euro-zone debt crisis. To eliminate spurious correlation generated by holidays, we eliminate the observations for which at least one holiday in a country occurs. Thus, we obtain the same number of observations across countries of 3942. The first log-difference generates the daily oil price and the stock indices' returns: $y_t = \log(P_t/P_{t-1}) \times 100$, where $P_t$ reflects the daily closing price at the given time t.

Figure 2 depicts the trajectory of the daily prices and returns series over the sample period. The daily price series in Figure 2 shows that WTI oil prices display a considerable rise and reach a peak of nearly $145.29 per barrel in July 2008. This progressive and rapid increase in oil prices is interrupted by two sudden falls in late 2008 and from mid-2014. In fact, in 2008 the recent financial crisis originated from the U.S banking sector and spread to other markets, especially the oil market. After this date, oil prices experienced a sharp decline from nearly $150 per barrel to $33.87 per barrel by late 2008, and maintain a low level before rising again and reaching a high level around $110 per barrel between 2013 and mid-2014. However, since mid-2014, world oil prices displayed a second heavy fall and have continued to decline to this day. This second oil shock can be explained by the continued increase of oil supply by OPEC members, as well as, the decline of global economic activity resulting in stable but low levels of world-wide oil-demand.

For instance, the WTI price decreased to less than $32 in June 2014, and less than $26.55 in January 2016. In September 2018, the oil price increased again and reached a peak of nearly $75. The long period of oil price decline, coincided with the second oil price shock period, as mentioned before in this study. Furthermore, it can be seen from this figure that stock prices in oil-exporting countries behave similarly to oil price movement during the period of shocks. In fact, during the second oil price shock, a significant drop was registered in stock markets in oil-exporting countries. In contrast, the trajectories of stock prices in oil-importing countries differ from those of oil prices and oil-exporting countries.

Regarding the evolution of oil price and stock market returns, Figure 2 illustrates stylized facts, such as volatility clustering, for the oil and the six stock returns series for oil-exporting and oil-importing countries under study. Table 2 reports the descriptive statistics for oil and stock market indices prices and returns. The average returns are positive for all series. More precisely, with respect to the oil market, Russia and Norway hold the highest average returns, while the Japanese stock market index has the lowest average returns. The unconditional volatility, expressed by the sample's standard deviation, is similar for all stock markets. However, it is noticed that the Russian stock market registers the highest volatility, indicating that this market is mainly exposed to oil price stress than any other markets. Furthermore, negatives skewness values are common for all the series. Compared with the normal distribution, the highest values of kurtosis for all returns causes fat tails in the data series.

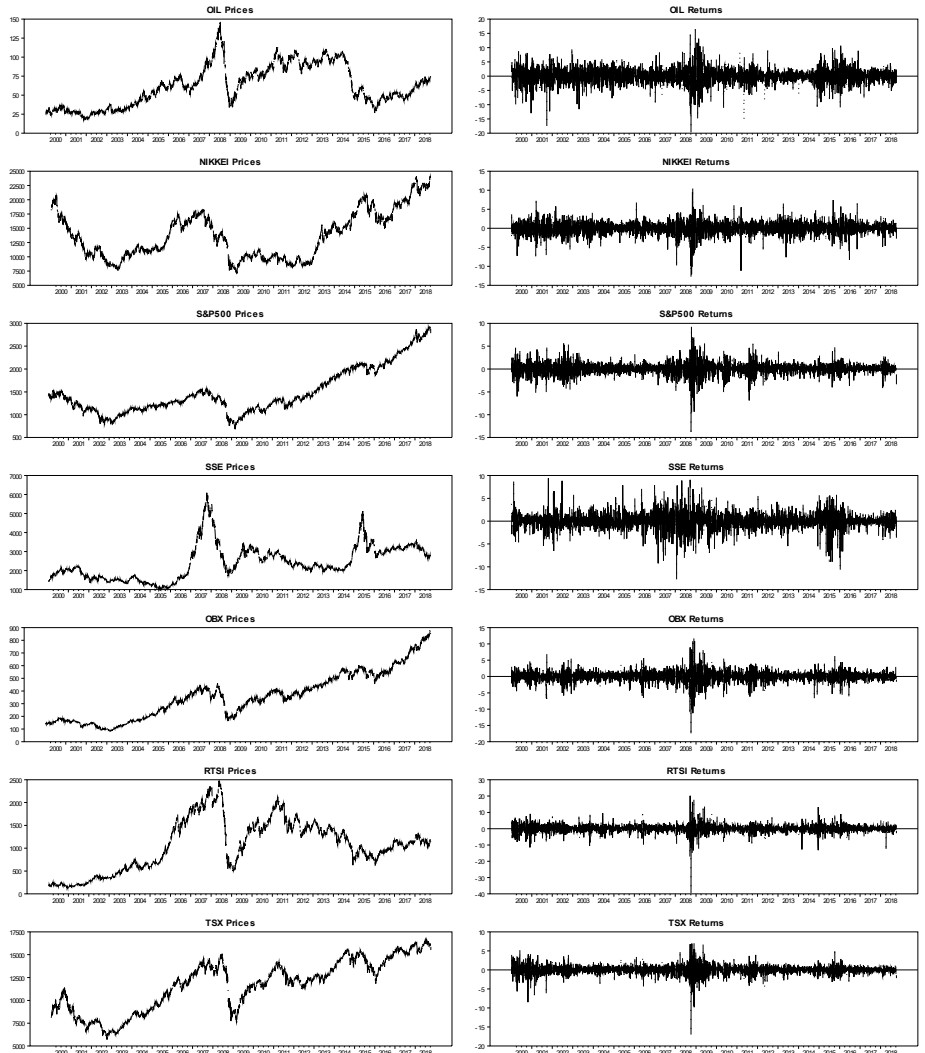

**Figure 2.** Evolution of daily prices and returns of WTI crude oil and stock markets for oil-exporting and oil-importing countries.

**Table 2.** Descriptive statistics of prices and returns series for oil and stocks.

|  | Mean | Median | Max | Minimum | St. Dev. | Skewness | Kurtosis | J-B | Probability |
|---|---|---|---|---|---|---|---|---|---|
| *Panel A: Prices series* | | | | | | | | | |
| WTI | 61.69 | 58.98 | 145.29 | 17.45 | 27.04 | 0.3721 | 2.1921 | 199.05 | 0.0000 |
| NIKKEI | 13.751.34 | 13.201.14 | 24.124.15 | 7054.98 | 4102.29 | 0.4814 | 2.1854 | 261.31 | 0.0000 |
| SP | 1487.01 | 1328.32 | 2930.75 | 682.55 | 497.69 | 1.0444 | 3.2665 | 724.36 | 0.0000 |
| SSE | 2415.84 | 2224.11 | 6092.06 | 1011.50 | 906.69 | 0.9684 | 4.1681 | 828.45 | 0.0000 |
| OBX | 356.44 | 348.50 | 881.01 | 83.13 | 185.12 | 0.5395 | 2.5721 | 213.79 | 0.0000 |
| RTSI | 1053.67 | 1065.14 | 2478.87 | 132.07 | 562.94 | 0.1594 | 2.1725 | 127.32 | 0.0000 |
| TSX | 11.639.69 | 12.110.90 | 16.567.40 | 5695.30 | 2805.34 | −0.2662 | 81.9420 | 230.68 | 0.0000 |
| *Panel B: Returns series* | | | | | | | | | |
| RWTI | 0.0273 | 0.1005 | 16.4097 | −19.6625 | 2.5692 | −0.3433 | 7.7938 | 3851.07 | 0.0000 |
| RNIKKEI | 0.0060 | 0.0369 | 10.4443 | −12.7154 | 1.6010 | −0.5664 | 8.8954 | 5917.99 | 0.0000 |
| RSP | 0.0174 | 0.0609 | 9.2407 | −13.7989 | 1.2856 | −0.5883 | 12.0179 | 13.581.0 | 0.0000 |
| RSSE | 0.0167 | 0.0112 | 9.4010 | −12.7636 | 1.7099 | −0.3516 | 8.4651 | 4985.59 | 0.0000 |
| ROBX | 0.0455 | 0.1160 | 11.6773 | −17.4087 | 1.6261 | −0.7273 | 12.6990 | 15.794.6 | 0.0000 |
| RRTSI | 0.0471 | 0.1405 | 20.2039 | −39.4545 | 2.4189 | −1.2075 | 27.2530 | 97.546.3 | 0.0000 |
| RTSX | 0.0164 | 0.0657 | 7.0040 | −16.9988 | 1.1648 | −1.2995 | 21.1148 | 54.993.5 | 0.0000 |

Notes: This table contains descriptive statistics (mean, maximum, minimum, standard deviation (St.Dev), skewness, kurtosis, and statistics of the Jarque-Bera test) for daily prices and returns series for oil and the six stock markets considered for the period running from 6 January 2000 to 15 October 2018. Skewness and kurtosis excess are presented. J-B is the Jarque-Bera normality test.

## 5. Results

### 5.1. Dynamic Conditional Correlation Estimation Results

Table 3 reports the parameter estimates of the bivariate DCC-FIGARCH model between oil and each stock market in oil-exporting and oil-importing countries. The results of diagonal tests show that there are no significant correlations or ARCH effect, which ensure that the model can secure residuals of conventional models.

**Table 3.** DCC-FIGARCH (1,1) parameter estimation results for the relationship between oil and stock markets.

| | Oil | Oil-Exporting Countries | | | Oil-Importing Countries | | |
|---|---|---|---|---|---|---|---|
| **Markets** | **WTI** | **RTS** | **TSX** | **OSEAX** | **S&P500** | **NIKKEI** | **SSE** |
| *Panel A: Mean equation* | | | | | | | |
| $c_1$ | 0.0482 | 0.1177 *** | 0.0473 *** | 0.1057 *** | 0.0625 *** | 0.0626 *** | 0.0266 * |
| | (1.397) | (3.840) | (3.655) | (5.914) | (4.415) | (2.952) | (1.198) |
| *Panel B: Variance equation* | | | | | | | |
| $\omega_i$ | 0.1528 | 0.2697 *** | 0.0219 *** | 0.0853 *** | 0.0428 *** | 0.0806 *** | 0.0065 ** |
| | (1.397) | (2.655) | (2.677) | (2.599) | (2.851) | (2.345) | (1.929) |
| $\alpha_1$ | 0.3064 *** | −0.019 *** | 0.1609 *** | 0.1251 * | 0.1022 | 0.1389 * | −0.1055 * |
| | (5.001) | (−0.2285) | (3.146) | (1.832) | (1.456) | (1.788) | (−1.804) |
| $\beta_1$ | 0.6656 *** | 0.4286 *** | 0.6144 *** | 0.4843 *** | 0.5135 *** | 0.5486 *** | 0.9693 *** |
| | (10.930) | (2.795) | (7.982) | (4.829) | (4.612) | (3.596) | (81.290) |
| $d$ | 0.4232 *** | 0.5038 *** | 0.5186 *** | 0.4656 *** | 0.5032 *** | 0.4933 *** | 1.1223 *** |
| | (7.385) | (4.650) | (6.326) | (6.020) | (5.530) | (4.081) | (18.810) |
| *Panel C: DCC (1,1) parameters* | | | | | | | |
| $a_1$ | — | 0.0165 *** | 0.0180 *** | 0.0160 *** | 0.0194 *** | 0.0613 ** | 0.0310 |
| | — | (5.259) | (4.310) | (4.713) | (5.570) | (1.779) | (0.499) |
| $b$ | — | 0.9797 *** | 0.9779 *** | 0.9805 *** | 0.9806 *** | 0.6121 | 0.8445 ** |
| | — | (248.300) | (186.700) | (238.100) | (274.700) | (1.318) | (2.109) |
| $a + b$ | — | 0.9963 | 0.9958 | 0.9965 | 1.0000 | 0.6734 | 0.8755 |
| $\rho_1$ | — | 0.4434 *** | 0.5392 *** | 0.5150 *** | −0.0718 * | 0.0770 *** | 0.0805 *** |
| | — | (6.625) | (9.056) | (8.043) | (−0.6166) | (4.323) | (4.652) |
| LL | — | −16.707.4 | −13.634.9 | −15.175.2 | −14.149.7 | −15.675.4 | −15.670.1 |
| *Panel D: diagnostic tests* | | | | | | | |
| Q(15) | 0.3435 *** | 0.4398 *** | 0.2396 *** | 0.8044 *** | 0.1143 *** | 0.8962 *** | 0.7145 *** |
| $Q_s$(15) | 0.9801 *** | 0.9712 *** | 0.7779 *** | 0.7677 *** | 0.9196 *** | 0.8288 *** | 0.7318 *** |
| Jarque-Bera test | 1833.5 *** | 3978.1 *** | 1275.8 *** | 1282.1 *** | 1014.7 *** | 873.16 *** | 2183.5 *** |

Note: This table provides the parameter estimation results of the DCC-FIGARCH(1,d,1) by the maximum likelihood method to assess the time-varying correlations between oil and stock markets of the selected countries. The Student's *t*-test of significance is reported in parentheses. LL is the likelihood value at the optimum. The superscripts (*), (**) and (***) indicate that the parameter is significant at 10%, 5%, and 1% levels, respectively.

Regarding the mean equation, the results show that the constant parameter is significant at 1% level, indicating that past returns significantly affect current market returns for all series. For the variance equation, the estimated results from Table 2 indicate that the long memory parameter ($d$) is statistically significant for all markets, indicating strong evidence of long memory[4] in the volatility of oil and stock return series. Furthermore, the ARCH and GARCH items, which capture shock dependence and volatility persistence, are always significant during the period of the study. This means that the current volatility of the return series is easily affected by the information in the previous period. Moreover, the relatively small values of ARCH coefficients ($\alpha$) indicate that conditional volatility does not change very rapidly under the impulses, but it tends to fluctuate gradually over time as suggested

---

[4]   The long memory in volatility implies that the volatility keeps in memory the consequences of shocks for a relative long period.

by the large magnitude of GARCH coefficients ($\beta$). The behavior of oil price volatility is typically similar to the patterns of stock market volatility.

For the DCC model, results reported in Table 3 show that the model is correct since $a + b < 1$. The significant value of the coefficient $a$ indicate that the volatility of recent market returns significantly influences the dynamic linkages between oil and the stock market in oil-exporting and oil-importing countries. Moreover, the great values of the coefficient $b$ (close to 1 in almost cases) show that the dynamic linkage between oil and stock markets in oil exporters and oil importers will continue for a long period of time. Our results extend those of (Wang et al. 2013) to some extent, who only used a Structural VAR analysis to investigate the effect of oil price shocks on stock market returns; however, we empirically confirm the long-term dynamic volatility linkages between oil and stock market returns in oil-exporting and oil-importing countries.

Figure 3 represents the time varying-correlations for oil-importing and oil-exporting countries. The time-varying correlations patterns indicate that the dynamic correlation is different to the net oil position of the country, even though, in most cases, positive values prevail.

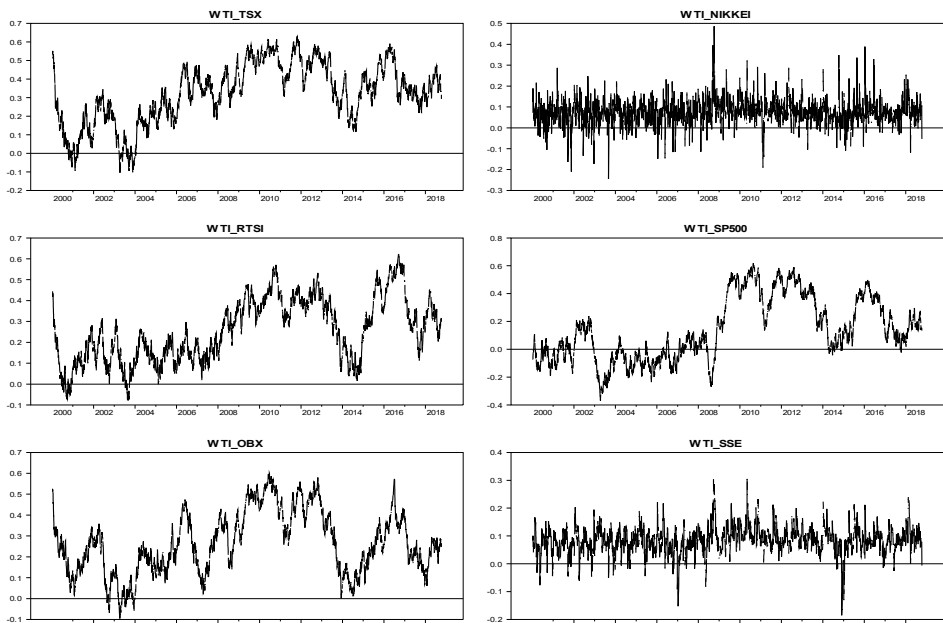

**Figure 3.** Time-varying correlation between WTI crude oil and stock market returns in oil-exporting and oil-importing countries.

### 5.1.1. The Case of Oil-Importing Countries

The time-varying correlations between the WTI and the U.S stock index reported in Figure 3 show that correlations move in tandem mostly in the positive region around a mean of zero with most modest variation. It is evident that dynamic time-varying correlations between oil prices and the U.S stock market have been changing from negative to positive since mid-2008. During the 2000–2001 period, which related to the dot-com bubble, the early-2000 recession in the U.S and the 9/11 terrorist attack; the correlations are strongly positive in the range of 0–0.1. In March 2003, during the second invasion of U.S troops in Iraq, the U.S stock markets exhibited negative correlations with oil prices. In fact, during the second war in Iraq, oil prices increased significantly, but, at the same time, caused the world stock market to react negatively. Then, the following period, running from 2006–2008, was characterized by a significant increase in oil prices due to rising demand, mainly due to Chinese economic development.

The correlation coefficient showed an increase and positive patterns ranging from −0.2 to 0.19. Since the global financial crisis of October 2008, which made world financial markets more interdependent, the dynamic correlations between oil price returns and stock markets have strengthened and moved

to the positive region in the range of 0.4–0.6. The positive correlation between oil and stock market prices can be explained by the fact that such a crisis caused stock markets to enter bearish territories and caused oil prices to also decrease heavily (Filis et al. 2011). The following period, from 2011–2014, which is associated with continuing unrest in the Middle East, reveals strong positive correlations between oil and U.S stock prices, except for the drop to −0.01 at the end of 2014 associated with the drop in the oil prices from $100 to $40. Time-varying correlations were positive and relatively high, ranging from 0.1–0.48 for the end of the sample. Moreover, the continuing decline in oil prices until 2016 was part of a particular geopolitical and economic context: the slowdown in the Chinese economy, the decline in global demand, Iran's return to international trade, the rivalry between the United States and Saudi Arabia for control of markets, etc. Given the place of oil in the global economy and its geostrategic importance, such a fall has many consequences.

Regarding the dynamic correlations between Chinese stock market and oil reported in Figure 3, we observe more volatile stock returns than in the U.S. The Chinese stock market is positively correlated to oil prices for most of the sample period and ranged between −0.2 and 0.31. Furthermore, the dynamic correlations seem less volatile during the first stage before the global financial crisis. The lowest negative correlation of −0.15 at the end of 2006 and the beginning of 2007 can be explained by the incredible growth that the Chinese economy was experiencing during this period. In fact, the Chinese SSE stock index was very volatile during this period due to the valuation methodology of the Chinese stock market, which was integrated with international standards for the first time and also due to a split share structure reform and sustainable growth of listed companies (Yao and Luo 2009; Cheng and Li 2014). After mid-2008, the dynamic correlations increased and moved in tandem with global financial and economic events. This is revealed from the considerable high positive correlation of 0.32 at the end of 2008, of 0.24 in the first quarter of 2018 and the bottom negative correlation of −0.2 in January 2015 when the global economy slowed. These results indicate that global economic and financial conditions influence the Chinese stock market. These findings are consistent with anterior literature (Broadstock et al. 2012; Boldanov et al. 2016).

Regarding the Japanese stock market, conclusions about the dynamic correlations are similar to those mentioned above. From Figure 3, we can see that the Japanese stock index (NIKKEI) and WTI are positively correlated for most of the period of study and balanced between the range of −0.22 and 0.49. During the first period before the global financial crisis, the dynamic correlations were always positive and reached a peak of 0.3. This period was characterized by different events, such as the terrorist attack in September 2001 and the U.S invasion in Iraq in 2003, which reveal weak positive correlations compared to the U.S market. The bottom negative correlation of −0.22 observed at the end of 2004 is due to the major events at this time, such as the highest activity of the Atlantic hurricanes. In fact, despite the uncertainty that the hurricanes created in the oil market, stock markets remained relatively calm. Additionally, the negative correlation of −0.18 observed in January 2011, can be explained by the continued conflicts in the Middle East which caused a precautionary demand shock in oil prices. These were cases of uncertainty-bearing shocks in which stock markets are expected to exhibit substantial increases in volatility. Furthermore, the spectacular high positive correlation of 0.48 in mid−2008 and of 0.4 at the beginning of 2016 are similar to those observed for the Chinese stock market and are related to the same world economic and financial events.

### 5.1.2. The Case of Oil-Exporting Countries

In this section, we move to the dynamic correlation between oil prices and stock market returns in oil-exporting countries. From Figure 3 we can observe that the dynamic correlations between oil and stock markets in oil-exporting and oil-importing countries are positive for most of the sample period. This can be explained by the fact that global economic activity during this period drives oil and stock prices to move in the same direction. We also observe that all oil-exporting stock markets are positively correlated to oil prices most of the time and the strongest average correlation is registered for the Canadian market, with a peak of 0.6 at the end of 2010 and 2011. For the Russian stock market,

we register a level of correlation of 0.6 during the end of 2010 and 2016. Finally, the smallest correlation level is registered for the Norwegian stock market. The bottom negative correlations of −0.22 and −0.15 between oil and Canadian and Russian stock market returns, respectively, is observed at the end of 2000, except for the Norwegian market for which we register a weak positive correlation close to 0. However, the 9/11 terrorist attack that took place at the World Trade Center (WTC) has moved correlation to the positive side for all oil-exporting countries, and the correlation has increased significantly for these countries.

Furthermore, we can observe that during the second war in Iraq which started in March 2003 all stock markets were exhibiting weak negative correlations with oil prices. In fact, during this period oil prices increased significantly, causing world stock markets to react negatively (Filis et al. 2011). Then for the following period between 2006 and before mid-2008, time-varying correlations moved to positive patterns and increased significantly for all oil-exporting countries as a result of oil price increases due to rising demand mainly in China. This shock considered as an aggregate demand-side oil price shock was expected to positively affect stock markets both in oil-importing and oil-exporting countries, as it reveals an increase in world trade mainly dominated by China. These findings are in harmony with prior literature suggesting that aggregate demand-side oil price shocks, originated by world economic growth, have a positive impact on stock prices (Hamilton 2009b; Kilian and Park 2009). After mid-2008, during the global financial crisis, world financial markets became more interdependent, and the dynamic correlations between oil price returns and stock market returns in oil-exporting countries strengthened and changed to a positive sign again for all markets, oscillating between 0.6 and 0.58. A plausible explanation for the observed positive correlations is that such a crisis caused the stock market to enter a bearish period, as mentioned above, and also caused oil prices to decline. The sudden drop in the dynamic correlation in December 2014 registered for all oil-exporting countries ranging from 0.1 for the Canadian market and close to 0 for the Russian and Norwegian markets is explained by the remarkable drop in oil prices at the end of the year 2014 from $100 to $40 per barrel.

For the following period running from 2015 to the end of our sample, we remark that the time-varying correlations for oil-exporting countries increased significantly and maintained their positive values, reaching a peak of 0.6 for the Russian market and more than 0.55 for the Canadian and the Norwegian markets. Overall, it is clear that oil-stock correlation, whatever the position of the economy (oil-importer or oil-exporter), is determined by financial and economic events (economic recession, financial crisis, stock market drop, etc.). This behavior is consistent with the information recently provided in a report of the EIA on what drives crude oil prices (7 February 2017).

### 5.1.3. Correlations between Stock Markets

As in the case of the oil-stock market's relationship, we employ the same model (DCC-FIGARCH) to assess the time-varying correlations between oil-importing and oil-exporting countries. The parameter estimation results are provided in Table 4, while Figure 4 plots the evolution of the correlation parameter over time.

**Table 4.** DCC-FIGARCH (1,1) parameter estimation results for the relationship between stock markets.

| Markets | TSX-NIK | TSX-SP | TSX-SSE | RTSI-NIK | RTSI-SP | RTSI-SSE | OBX-NIK | OBX-SP | OBX-SSE |
|---|---|---|---|---|---|---|---|---|---|
| $a_1$ | 0.2275 *** | 0.4716 * | 0.0088 | 0.2564 *** | 0.1374 | −0.0686 | 0.3241 *** | 0.2338 | 0.0084 |
| | (14.310) | (1.725) | (0.108) | (11.780) | (1.317) | (−0.579) | (15.500) | (1.215) | (0.112) |
| $b$ | $1.5 \times 10^{-14}$ | 0.0234 *** | 0.0019 * | 0.0170 ** | 0.0071 *** | 0.0041 *** | 0.0078 * | 0.0107 *** | 0.0020 ** |
| | (0.977) | (5.509) | (1.812) | (1.962) | (3.910) | (2.653) | (1.858) | (4.493) | (2.111) |
| $a + b$ | 0.2275 | 0.4949 | 0.0107 | 0.2733 | 0.1445 | −0.0645 | 0.3319 | 0.2445 | 0.0105 |
| $\rho_1$ | 0.7982 *** | 0.9756 *** | 0.9977 *** | 0.9379 *** | 0.9927 *** | 0.9956 *** | 0.9679 *** | 0.9889 *** | 0.9980 *** |
| | (0.093) | (194.90) | (741.90) | (24.06) | (507.80) | (565.80) | (49.63) | (391.30) | (841.30) |

Note: This table provides the parameters estimation results of the DCC-FIGARCH(1,d,1) by the maximum likelihood method to assess the time-varying correlations between oil-importing and oil-exporting stock markets. The Student's *t*-test of significance is reported in parentheses. The superscripts (*), (**), and (***) indicate that the parameter is significant at 10%, 5%, and 1% levels, respectively.

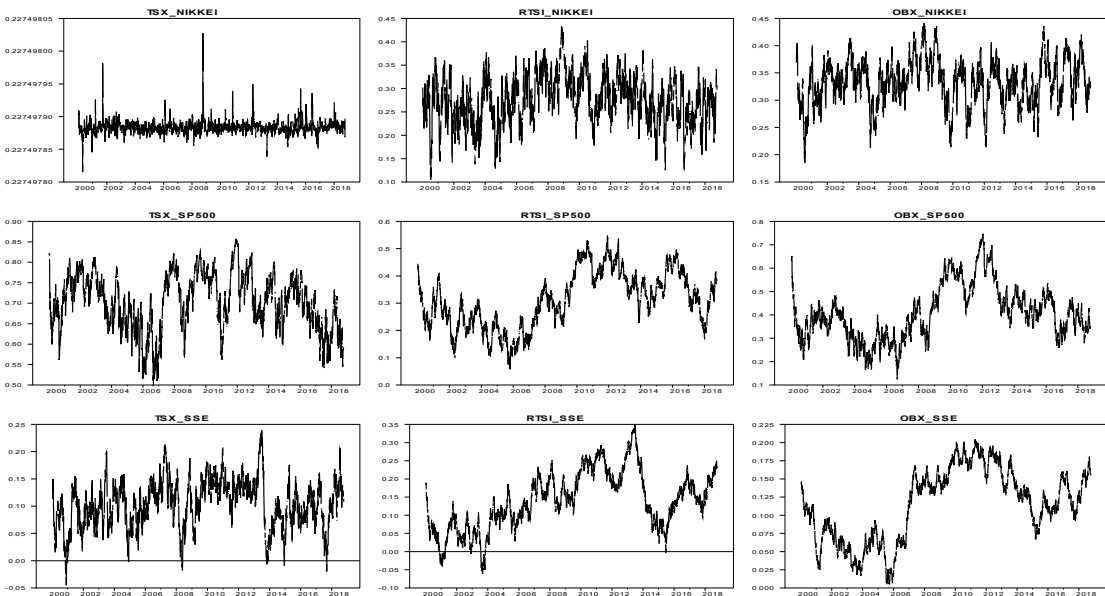

**Figure 4.** Time-varying correlation between stock market returns of oil-exporting and oil-importing countries.

Panel B of Table 5 reports the descriptive statistics of the dynamic time-varying correlations between stock markets in net oil-exporting and oil-importing countries. From these statistics, we can observe that the dynamic correlations vary between 0.10 and 0.70. The highest correlation level is registered between the Canadian and American markets, while the smallest is observed for the Canadian-Chinese pair. Regarding the standard deviation of the time-varying correlations between stock markets, results emphasize that the highest variability of correlations is associated with the OBX-S&P500 pair, followed by the S&P500-RTSI relationship, indicating that the degree of integration is unstable between these markets. Moreover, other pairs display low correlation variability, with the minimum between the Canadian and Japanese markets.

**Table 5.** Descriptive statistics of the dynamic conditional correlations (DCC).

| | Mean | Median | Maximum | Minimum | St. Dev. | Skewness | Kurtosis |
|---|---|---|---|---|---|---|---|
| *Panel A: DCC between oil and stock markets* | | | | | | | |
| WTI_TSX | 0.3202 | 0.3300 | 0.6299 | −0.1039 | 0.1630 | −0.3606 | 2.4435 |
| WTI_RTSI | 0.2537 | 0.2425 | 0.6222 | −0.0781 | 0.1520 | 0.1714 | 2.1309 |
| WTI_OBX | 0.2636 | 0.2481 | 0.6016 | −0.0955 | 0.1503 | 0.1802 | 2.2294 |
| WTI_SP | 0.0748 | 0.0784 | 0.1335 | 0.0110 | 0.0440 | −0.1014 | 1.4813 |
| WTI_NIKKEI | 0.1391 | 0.0942 | 0.6188 | −0.3696 | 0.2277 | 0.3106 | 2.0126 |
| WTI_SSE | 0.0798 | 0.0818 | 0.1090 | 0.0470 | 0.0223 | −0.1164 | 1.4686 |
| *Panel B: DCC between oil-importing and oil-exporting stock markets* | | | | | | | |
| TSX_NIKKEI | 0.2364 | 0.2364 | 0.2364 | 0.2364 | 0.0214 | −0.1436 | 3.0340 |
| TSX_SP | 0.7021 | 0.7060 | 0.8571 | 0.5004 | 0.0686 | −0.3202 | 2.5761 |
| TSX_SSE | 0.1030 | 0.1057 | 0.2396 | −0.0449 | 0.0436 | −0.1389 | 3.1060 |
| RTSI_NIKKEI | 0.2751 | 0.2772 | 0.4332 | 0.1044 | 0.0518 | −0.1319 | 2.9275 |
| RTSI_SP | 0.3087 | 0.3079 | 0.5475 | 0.0575 | 0.1032 | 0.0130 | 2.1524 |
| RTSI_SSE | 0.1381 | 0.1355 | 0.3475 | −0.0607 | 0.0801 | 0.0660 | 2.4806 |
| OBX_NIKKEI | 0.3325 | 0.3345 | 0.4415 | 0.1856 | 0.0420 | −0.2587 | 3.0610 |
| OBX_SP | 0.4065 | 0.3973 | 0.7472 | 0.1243 | 0.1169 | 0.4672 | 2.7932 |
| OBX_SSE | 0.1137 | 0.1206 | 0.2046 | 0.0043 | 0.0497 | −0.2338 | 1.9670 |

Note: Descriptive statistics correspond to daily time-varying correlations between oil prices and the six stock indices and time-varying correlations between stock market returns for the period from January 2000 to October 2018.

Figure 4 reports the patterns of the dynamic correlations between stock markets during the sample period. We remark that for all considered pairs the behavior of the correlation is generally similar. Further, we observe that all stock market pairs are positively correlated, expect for these Chinese market associations with the Canadian and Russian markets, for which correlations oscillate between positive and negative values. In most cases, a downward trend in the correlation level is observed during the first period of our sample. Moreover, an increasing pattern between 2001 and mid-2008 is observed for all stock market pairs, except for the Canadian-Chinese pair, for which the dynamic correlation turned to negative with weak levels close to 0. For the Norwegian-American stock market correlations, we observe a small decline in the correlation. In fact, during the period of crisis, stock markets became more integrated and highly correlated. Having examined the time-varying correlations between stock market returns, investigation of the effect of oil-stock correlation in these correlations is continued.

### 5.2. The Role of Oil Prices in Driving Correlation between Stock Markets

In this paper, we aim to investigate the role of crude oil prices in determining correlations between stock market returns among oil-importing and oil-exporting economies through oil-stock correlations in each country. To do so, we use the regression models described in Equations (12) and (13) to examine whether the degree of linkage of a stock market to oil could drive stock markets correlations (as shown in Figure 1). Table 6 provides the parameter estimation of both static and regime-switching regression models described by Equations (12) and (13), respectively.

**Table 6.** Parameter estimation results of the static and switching-regime regression models.

| E,I | TSX-NIKKEI | TSX-SP | TSX-SSE | RTSI-NIKKEI | RTSI-SP | RTSI-SSE | OBX-NIKKEI | OBX-SP | OBX-SSE |
|---|---|---|---|---|---|---|---|---|---|
| *Panel A: Static model* | | | | | | | | | |
| $\alpha_0$ | 0.2275 *** | 0.7240 *** | 0.0582 *** | 0.2506 *** | 0.2386 *** | 0.0512 *** | 0.3264 *** | 0.3479 *** | 0.0548 *** |
| | $(7.0 \times 10^8)$ | (281.761) | (34.446) | (137.424) | (107.665) | (20.460) | (211.408) | (130.664) | (34.537) |
| $\alpha_1$ | 0.0000 | −0.1615 *** | 0.0924 *** | 0.0505 *** | 0.1122 *** | 0.3082 *** | −0.0117 *** | 0.0187 | 0.1949 *** |
| | (−0.271) | (−16.933) | (23.558) | (9.448) | (10.827) | (44.727) | (−2.634) | (1.622) | (45.898) |
| $\alpha_2$ | $6.6 \times 10^{-8}$ *** | 0.2146 *** | 0.1808 *** | 0.1562 *** | 0.2996 *** | 0.1040 *** | 0.1226 *** | 0.3857 *** | 0.0897 *** |
| | (25.987) | (31.430) | (12.937) | (10.170) | (43.300) | (4.544) | (9.700) | (50.736) | (6.432) |
| $R^2$ | 0.1489 | 0.2252 | 0.1777 | 0.0544 | 0.6333 | 0.3620 | 0.0237 | 0.5917 | 0.3738 |
| F-stat | 344.53 | 572.14 | 425.36 | 113.35 | 3400.61 | 1117.17 | 47.86 | 2853.56 | 1175.45 |
| Prob | 0.0000 | 0.0000 | 0.0000 | 0.0000 | 0.0000 | 0.0000 | 0.0000 | 0.0000 | 0.0000 |
| *Panel B: Regime switching model* | | | | | | | | | |
| $\alpha_{0,1}$ | 0.2275 *** | 0.6742 *** | 0.0229 *** | 0.2956*** | 0.2985 *** | 0.0300 *** | 0.3064 *** | 0.3973 *** | 0.0346 *** |
| | (8.214) | (256.249) | (10.188) | (162.149) | (111.815) | (15.559) | (156.664) | (143.628) | (24.492) |
| $\alpha_{1,1}$ | $1.5 \times 10^{-9}$ | −0.1686 *** | 0.0872 *** | 0.0142 *** | 0.0532 *** | 0.2681 *** | −0.0593 *** | 0.0888 *** | 0.1129 *** |
| | (0.124) | (−19.228) | (16.329) | (2.576) | (5.357) | (50.475) | (−11.043) | (7.816) | (26.418) |
| $\alpha_{2,1}$ | $7.1 \times 10^{-8}$ | 0.1960 *** | 0.1963 *** | 0.1532 *** | 0.2798 *** | 0.0568 *** | 0.0898 *** | 0.3167 *** | 0.0845 *** |
| | $(0.2 \times 10^{-7})$ | (31.838) | (11.663) | (11.335) | (48.347) | (3.022) | (6.211) | (39.801) | (6.325) |
| $\alpha_{0,2}$ | 0.2275 | 0.7673 *** | 0.1005 *** | 0.2122 *** | 0.1938 *** | 0.1556 *** | 0.3619 *** | 0.3159 *** | 0.1088 *** |
| | $(2.7 \times 10^{-6})$ | (311.381) | (59.673) | (107.045) | (56.697) | (45.425) | (204.084) | (89.178) | (89.409) |
| $\alpha_{1,2}$ | $−1.1 \times 10^{-9}$ | −0.1425 *** | 0.0634 *** | 0.0453 *** | 0.0010 *** | 0.2645 *** | −0.0156 *** | −0.0946 *** | 0.1087 *** |
| | $(−3.4 \times 10^{-4})$ | (−15.096) | (15.303) | (7.763) | (0.070) | (23.551) | (−3.426) | (−7.028) | (35.977) |
| $\alpha_{2,2}$ | $7.0 \times 10^{-8}$ | 0.1812 *** | 0.0980 *** | 0.0975 *** | 0.3563 *** | −0.1402 *** | 0.0657 *** | 0.3817 | 0.0778 *** |
| | (1.002) | (26.124) | (6.737) | (5.896) | (46.669) | (−4.807) | (5.610) | (49.566) | (8.338) |
| $Log(\sigma_\varepsilon^2)$ | −17.272 *** | −3.3510 *** | −3.6786 *** | −3.4643 *** | −3.3529 *** | −3.2182 *** | −3.6485 *** | −3.1164 *** | −3.8812 *** |
| | (−5.245) | (−291.711) | (−317.798) | (−297.137) | (−291.742) | (−283.702) | (−315.980) | (−272.668) | (−341.963) |
| $P_{11}$ | 0.1189 *** | 4.1521 *** | 3.7582 *** | 3.7620 *** | 5.0579 *** | 5.4901 *** | 3.9231 *** | 4.6228 *** | 5.7099 *** |
| | (2.841) | (20.684) | (21.595) | (24.901) | (18.734) | (17.874) | (19.964) | (19.964) | (13.094) |
| $P_{21}$ | −0.0604 *** | −4.3542 *** | −4.1308 *** | −3.5205 *** | −4.5554 *** | −4.8179 *** | −4.0604 *** | −4.5695 *** | −6.2004 *** |
| | (−2.333) | (−21.814) | (−23.435) | (−23.560) | (−17.111) | (−15.464) | (−22.900) | (−19.438) | (−13.143) |
| LL | 64.310.8 | 7423.2 | 8686.0 | 7769.0 | 7502.0 | 7005.2 | 8567.2 | 6547.7 | 9654.4 |

Note: This table contains the parameter estimates of both static and regime switching models given in Equations (12) and (13), for oil-exporting and oil-importing stock market pairs. The Student's *t*-statistics are reported in parentheses. LL is the log-likelihood value at the optimum and. R2 is the regression coefficient. $P_{11}$ and $P_{21}$ are the estimated transition parameters of regime 1 and regime 2 respectively. The superscripts (***) indicate that the parameter is significant at 10%, 5% and 1% levels, respectively.

Estimation results show that the parameter $\alpha_1$, representing the effect of the time-varying correlation between oil and stock market returns in oil-exporting countries in driving stock markets correlations, is significant at the 1% level for all stock pairs except for the TSX-NIKKEI pair. The parameter $\alpha_2$, explaining the effect of the dynamic correlations between oil and stock market returns in oil-importing countries in deriving stock market correlations, is always significant at the 1% level for

all stock markets pairs. This results show that oil-stock correlations play an important role in driving correlations between oil-importing and oil-exporting countries stock market returns. Furthermore, we observe a positive effect of the degree of linkage to oil on the stock markets relationship for all pairs, except of some cases including oil-exporter countries. Additionally, the F-statistic emphasizes that all estimated regressions are globally significant.

To provide more information about the role of oil prices in driving the relationships between oil-importing and oil-exporting countries, we estimated the same regression model in a regime-switching perspective in which the study is reexamined by taking account different correlation levels with oil. The estimation of this model by the maximum likelihood method is reported in panel B of Table 6. Results suggest that different regression and transition parameters in different regimes are statistically significant at the 1% level, indicating that the investigated effect of the oil market depends on the low or high correlation regimes. Furthermore, we generally observe a positive effect of the degree of linkage to oil in both low and high correlation regimes except for some cases related to oil-exporting countries. Thus, in both static and regime switching perspectives, we could conclude that oil prices represent a channel through which the linkage between oil-related countries occurs.

## 6. Policy Implication

Findings from this research have potentially important implications for investors, portfolio management and policy-makers, and also lead to greater clarification of oil price pressures on stock market linkages. Moreover, findings highlight the difference between calm and turbulent periods. Furthermore, greater insight has been shed on the role of crude oil as an important financial tool for asset allocation and risk management for different markets. In fact, our findings have potential benefits for investors and financial market players in terms of risk management. First, investors can use our finding for hedging purposes, for example, in calm periods, oil-stock correlations are usually weak and, thus, oil can be used by investors as a hedge against stock market volatility. Second, since the research documented a strong significant and time-varying correlation between oil and stock markets, long-short strategies can be effectively used. Thus, investors can adjust their portfolio's exposure to risk. Third, based on negative and positive correlations with oil market, long-short hedging strategies among stock markets for oil-related countries can be more effectively designed, by identifying the forces of oil price shocks and the position of the country in the global oil market. Finally, our findings highlight that, in the most cases the oil-stock correlation is positive with different magnitude. This finding supports the assumption of anterior research suggesting that the speculation and financialization of the oil market has resulted in an increased oil-stock correlation. This holds mainly for oil-importing economies, as these countries exhibit mainly positive correlation with oil market for different time period. Consequently oil is no longer attractive as a hedge for investors operating in stock markets of oil-importing countries.

Moreover, the findings imply that signals from the oil market can have predictive power over the relationship between stock markets in oil-importing and oil-exporting countries. The relationship between these markets is sensitive to oil price shocks. This result may be explained by the fact that the trading behavior of investors is related to the movement of oil prices, which affects the investors' behavior in stock markets. Therefore, the interaction between these stock markets and oil prices suggest that policymakers can supplement their market monitoring mechanisms and investment activity with alternative factors related to oil markets such as oil price uncertainty, volatility, and oil-stock correlations. In this way, the oil-stock linkage could be a crucial factor, which can be used with other factors, affecting the relationship between stock markets in oil-importing and oil-exporting countries.

## 7. Conclusions

The objectives of this study are, first, to examine the conditional time-varying correlations between oil price shocks and stock market returns in major oil-exporting and oil-importing countries. Second, we aim to analyze the role of oil in driving dynamic correlation between stock market returns in these

countries as shown in Figure 1. This paper integrates the two streams of literature: the relationship between oil prices and stock prices and the relationship between stock markets using oil price shocks as a major mediator in driving the linkages between stock markets. Furthermore, this study investigates the above relationships in such a comparative context between major oil-exporting and oil-importing economies in the world. Finally, this paper explores the oil-stock linkages using the dynamic conditional correlation (DCC)-FIGARCH model. Then, we aim to investigate the role of oil prices in deriving the dynamic correlation between stock markets in these countries. To the author's best knowledge, this is the first paper to make such an attempt. In doing so, it is possible to best understand the dynamic relationships between oil price changes and stock market returns and the relationships between stock markets in these countries regarding their reliance on oil.

Overall, our findings show that the responses of stock market indices to oil price shocks are time-varying and oscillate between positive and negative values, although positive values prevails for both oil-exporting and oil-importing countries. Moreover, the impact of oil price shocks on stock market returns for oil-exporting countries is much higher than for oil-importing countries. These results suggest that the stock markets in oil-exporting economies are more susceptible to oil price shocks. Regarding, oil-importing countries, results show that all stock markets respond in the same way to oil price shocks but with different magnitudes. In major cases, oil prices positively impact stock markets in the U.S and Japan, while the Chinese stock market responded mostly positively with more volatility, especially after the 2008 global financial crisis. This result indicates that China is more integrated with the world economy and events than ever before.

Additionally, our results show that the responses of stock markets to oil price shocks depend not only on the net position of the country in the global oil market but also on general forces deriving oil prices to change such as wars, geopolitical and economic events, financial crises, and changes in the global business cycle. Our findings partly agree with previous studies, such as those conducted by (Bhar and Nikolova 2010); (Filis et al. 2011); (Arouri et al. 2011); and (Wang et al. 2013), among others, which report a time-varying relationship between oil price shocks and stock markets. Nevertheless, the findings contrast anterior empirical results that suggest that there is no relationship between oil price changes and stock markets (Apergis and Miller 2009; Reboredo and Rivera-Castro 2014), among others.

In the second part of this research, we have analyzed the role of oil-stock dynamic correlations in deriving relationships between stock markets in oil-related countries. For this, we used a static and regime-switching regression models to investigate the impact of high and low-correlation regimes on stock markets linkages. Our results show that oil-stock conditional correlations significantly impact stock market linkages in these countries. Additionally, we find evidence of positive effects of the degree of linkage to oil in both high and low correlation regimes. These findings indicate that the oil market is positioned as a mediator between these stock markets. This result suggests that policymakers can, in fact, utilize oil prices in order to model and monitor interrelations between stock markets.

Finally, we encourage future research to extend the literature in many ways. First, it is interesting to examine oil-stock relationships for non-energy stocks in order to identify whether energy-related stocks do not drive the evident oil-stock co-movement. Second, it is possible to examine the oil-stock relationships using more sophisticated approach, such as the copula theory and the quantile regression approach during bullish and bearish periods, and under different volatility regimes, to improve the modeling of oil-stock linkages and of the role of oil price shocks in predicting interrelation between stock markets. Finally, a decomposition of the period of study to normal and turmoil periods could also be considered in order to test the contribution of oil prices in determining the stock market relationships during every sub-sample.

**Author Contributions:** Both authors contributed equally to this research.

**Funding:** The authors gratefully acknowledge the approval and the support of this research study by the grant no 7435-BA-2017-1-8-F from the Deanship of Scientific Research at Northern Border University, Arar, K.S.A.

**Conflicts of Interest:** The authors declare no conflict of interest.

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
