# Peer review of "Do Crude Oil Prices Drive the Relationship between Stock Markets of Oil-Importing and Oil-Exporting Countries?"

_economies, doi:10.3390/economies7030070_

Round 1

Reviewer 1 Report

The abstract should offer a clear view in terms of the contribution of the paper, rather than indicating that the research “tries to investigate”, this is not a good start to the discussion. Oil driven economies will be affected by the oil market as it has been clearly discussed by existing research in the area, as such, the initial points should clarify the new insights from the paper. Maybe a different methodology, countries that have not been considered before, specific time periods that have not been analysed…?

The introduction emphasises on the use of the DCC-GARCH model to examine time varying correlation patters. As the paper is supported by the DCC-FIGARCH insights on the main motivation to select this type of model is needed. The discussions indicate that oil-exporting and oil-importing countries would be analysed, however, there is no clarity in terms of the countries that are involved, and this could be clearly presented in a relevant table that should be integrated around line 54 that is where the authors are making this point. The authors are considering policy making issues and also portfolio diversification strategies. The authors would be better off to align their discussions to a particular area of interest that would help them to offer a more focused analysis.

The paper main contribution seems to be the implementation of a DCC-FIGARCH model to support the analysis. It is not clear, why this model outcomes should be considered in the context of existing research, and what the authors’ main arguments are in terms of its value, and contribution to existing research in the area.

The introduction is too long and disorganised. The authors need to offer a clear view on the main motivation of the paper and its value around the extant literature and clarify why this paper is needed. The introduction would benefit from a careful review that allow the authors communicating the purpose of the research and its justification in a more efficient manner.

The literature review could be organised around research studies that examine oil-importing countries major outcomes and the same could be done in terms of oil-exporting economies. At the moment, the discussions are mixed, and there is no clarity on why the two type of economies need to be differentiated and what are the common trends among them. The reader would gain a better understanding of existing research core contributions if the two types of countries are considered separately and if insights in terms of the core elements that characterise research in the field and that help the authors to consider their own research contribution are considered. Figure one (line 158, page 4) requires integration throughout the discussions and the source of information should be clearly identified. Overall, the literature review section lacks structure and the clear presentation of arguments that justify the research paper and how it is defined by existing research in the field. Additionally, the paper requires a clear assessment of the core research findings and how they are considered to develop the research context that supports the paper.

The methodological research framework required clarification in terms of the benefits of using a DCC-FIGARCH when compared to other research methodologies, and clarifying why a student-t distribution was the chosen distribution when considering other options. Table 1 should clear identify the variables under consideration and explain clearly the outcomes when comparing prices and returns for the series under study and their implications.

The paper does not present a section that clearly explains the data set and how it was selected. The authors offer some insights on the section dealing with the results, however, the data and research methodology should be presented before research outcomes are introduced so that the reader is able to gather all the basic points of the methodological research framework.

As per the results section, the paper does not examining all oil exporting and oil importing economies, and it is based on a very restrictive selection of countries. This is a point that needs to be considered, as the paper title is misleading, and also early discussions presented in the abstract and the introduction. The data section should offer all details regarding data, its frequency, time period, etc., as mentioned above. The discussions need to clarify the importance of looking at periods of crisis and the implications for the study in terms of changes of patterns and the dynamic relationship exhibited by the involved markets.  Line 264 needs to consider the presentation of the formula to align it to conventional presentation in terms of returns as per the literature.

In line 305 the authors indicate that the constant parameter is significant and one percent level, but what is the interpretation of such an outcomes? The discussions that followed does not offer a clear examination of the outcomes and the main implications in terms of the implemented models and how the results should be considered in the context of the reviewed literature.

The section examining research findings should offer a clear view in terms of the major contribution of the paper and the results implications and how they should be considered.

The conclusion section is too long and it should be reconsidered to ensure that the core research outcomes, the main contributions and the value of the conducted study are clearly communicated to the reader.

The authors should consider professional proof reading the paper to ensure that typos, grammar mistakes, etc., are dealt with.

Author Response

rather than indicating that the research “tries to investigate”, this is not a good start to the discussion. Oil driven economies will be affected by the oil market as it has been clearly discussed by existing research in the area, as such, the initial points should clarify the new insights from the paper. Maybe a different methodology, countries that have not been considered before, specific time periods that have not been analysed…?

Response 1: the abstract has been reformulated.

Point 2: The introduction emphasises on the use of the DCC-GARCH model to examine time varying correlation patters. As the paper is supported by the DCC-FIGARCH insights on the main motivation to select this type of model is needed. The discussions indicate that oil-exporting and oil-importing countries would be analysed, however, there is no clarity in terms of the countries that are involved, and this could be clearly presented in a relevant table that should be integrated around line 54 that is where the authors are making this point. The authors are considering policy making issues and also portfolio diversification strategies. The authors would be better off to align their discussions to a particular area of interest that would help them to offer a more focused analysis.

Response 2: insights of the motivation to select the DCC-FIGARCH model are cited in the methodology section. Moreover we added a footnote in the introduction section to explain the choice of the FIGARCH model.

The table introducing the chosen countries under study is integrated.

Point 3: The paper main contribution seems to be the implementation of a DCC-FIGARCH model to support the analysis. It is not clear, why this model outcomes should be considered in the context of existing research, and what the authors’ main arguments are in terms of its value, and contribution to existing research in the area.

Response 3: the DCC-GARCH model early implemented by Engle (2002) has been largely used in the existing literature to investigate dynamic linkage between financial markets. According to Engle (2002), this model provides a very good approximation to a variety of time-varying correlation processes. Additionally, the comparison of DCC with simple multivariate GARCH and several other estimators shows that the DCC is often the most accurate. furthermore, several studies in the literature provide indisputable evidence that oil and stock markets do present dynamic interrelation ( see i.e.,  Ratti 2009; Reboredo 2010; Filis, Degiannakis, and Floros 2011; Daskalaki and Skiadopoulos 2011; Chang and Yu 2013; Reboredo and Rivera-Castro 2014; Zhang and Li 2014, Boldanov et al., 2016, Zhu et al., 2016; Aydogan et al., 2017). Thus the oil-stock relationship could present heterogeneous behaviour at different time, and thus it should be investigated with a dynamic approach. Moreover the extention of the GARCH specification to the FIGARCH is justified with the presence of the long memory phenomenon frequently observed in financial time series. So that, the DCC-FIGARCH approach adopted in this study is more appropriate to investigate jointly, the time-varying correlation patterns between oil and stock markets and between oil-related stock markets, and to consider for different financial stylized fact characterising the return series of oil and financial markets.

Point 4: The introduction is too long and disorganised. The authors need to offer a clear view on the main motivation of the paper and its value around the extant literature and clarify why this paper is needed. The introduction would benefit from a careful review that allow the authors communicating the purpose of the research and its justification in a more efficient manner.

Response 4: the introduction has been rewritten and reorganized.

Point 5: The literature review could be organised around research studies that examine oil-importing countries major outcomes and the same could be done in terms of oil-exporting economies. At the moment, the discussions are mixed, and there is no clarity on why the two type of economies need to be differentiated and what are the common trends among them. The reader would gain a better understanding of existing research core contributions if the two types of countries are considered separately and if insights in terms of the core elements that characterise research in the field and that help the authors to consider their own research contribution are considered. Figure one (line 158, page 4) requires integration throughout the discussions and the source of information should be clearly identified. Overall, the literature review section lacks structure and the clear presentation of arguments that justify the research paper and how it is defined by existing research in the field. Additionally, the paper requires a clear assessment of the core research findings and how they are considered to develop the research context that supports the paper.

Response 5: in the existing literature, the majority of studies have investigated the relationship between oil and stock markets in oil-importing and oil-exporting countries simultaneously. In fact, the main objective of these studies is to observe how stock markets response to oil shocks? More specifically, do stock market responses to oil price shocks vary depending on the net-oil position of the economy in the oil market? And how do oil-stock nexus is affected by economic and geopolitical uncertainty in oil exporter compared to oil importer economies. Thus, results from such comparative studies can help policy makers and local and international investors to better adjust their portfolio against different risk factors related to oil price changes.  In this context, few studies have analyzed the oil-stock nexus in the context of oil-exporting or oil-importing countries separately. As a result, the literature review section, in this study, was structured on the light of the findings of the existing studies. First part of the literature review deals with the different finding regarding the oil-stock nexus (positive, negative or no significant interrelation). The second part introduces the main research dealing with oil-stocks relationship in oil exporting and oil importing countries and the third part introduce same research that focused either on the oil-stock relationship in oil-exporter or oil-importer countries separately. The last part introduces the main studies dealing with the time-varying linkage between oil and stock markets, given that these studies are in a direct relation with the actual research.  

Figure one was designed by the authors to better explain the main objective and contribution of this research. It was integrated in the text.

Point 6: The methodological research framework required clarification in terms of the benefits of using a DCC-FIGARCH when compared to other research methodologies, and clarifying why a student-t distribution was the chosen distribution when considering other options. Table 1 should clear identify the variables under consideration and explain clearly the outcomes when comparing prices and returns for the series under study and their implications.

Response 6: the authors have added same clarification and arguments against the use of the DCC-FIGARCH methodology compared to other methodologies used in the same research area. The use of the student-t distribution is justified by the existence of excess kurtosis, and the results of the Jraque-Bera test leading to the rejection of the null hypothesis of normal distribution and confirm that return series under study exhibit fat tails, thus normal distribution cannot adequately fit our return series.

Point 7: The paper does not present a section that clearly explains the data set and how it was selected. The authors offer some insights on the section dealing with the results, however, the data and research methodology should be presented before research outcomes are introduced so that the reader is able to gather all the basic points of the methodological research framework.

Response 7: a section introducing the data and the countries selected is added to the manuscript.

Point 8: As per the results section, the paper does not examining all oil exporting and oil importing economies, and it is based on a very restrictive selection of countries. This is a point that needs to be considered, as the paper title is misleading, and also early discussions presented in the abstract and the introduction. The data section should offer all details regarding data, its frequency, time period, etc., as mentioned above. The discussions need to clarify the importance of looking at periods of crisis and the implications for the study in terms of changes of patterns and the dynamic relationship exhibited by the involved markets.  Line 264 needs to consider the presentation of the formula to align it to conventional presentation in terms of returns as per the literature.

Response 8: in the result section, we introduce finding form the methodology used and applied for the selected oil-importing and oil exporting countries. The section data represent all detail about the selected countries, reason of the choice of these countries, data frequency. The choice of the period under study is justified by the fact that this period includes different economic and geopolitical events. The purpose of choosing such period is to examine the different responses of oil-related countries to oil price shocks related to such events. Moreover we aim to compare how oil-stock interrelations are affected by different economic events and if the impact changes in oil-importers compared to oil exporters economies. The formula employed to calculate oil price and stock market returns was integrated.

Point 9: In line 305 the authors indicate that the constant parameter is significant and one percent level, but what is the interpretation of such an outcomes? The discussions that followed does not offer a clear examination of the outcomes and the main implications in terms of the implemented models and how the results should be considered in the context of the reviewed literature.

Response 9: the interpretation of the result was integrated on the text.

The main objective of using the DCC-FIGARCH model in this study is to examine the pattern of the dynamic conditional correlations between oil and stock market, and those between stock markets of oil-importing and oil exporting countries. To do this, we focus on the plots of dynamic correlations issued from the DCC model which give more detailed pictures of the patterns of oil-stocks correlations over the sample period considered. Thus, we can observe how oil-stock interrelations are affected by different economic and geopolitical events occurred during the period under study.

Point 10: The section examining research findings should offer a clear view in terms of the major contribution of the paper and the results implications and how they should be considered.

Response 10: we added a section for the policy implication to further clarify the major contribution of the research and how investors, portfolio managers and policymakers might benefit from our findings.

Point 11:  The conclusion section is too long and it should be reconsidered to ensure that the core research outcomes, the main contributions and the value of the conducted study are clearly communicated to the reader.

Response 11: the conclusion has been reorganized.

The added and modified text is written in red color.

Reviewer 2 Report

The authors investigate the role of the oil market in driving the dynamic relationship between stock markets in oil-related countries. They employ the DCC-FIGARCH model to characterize the dynamic relationship between these markets. They find the oil-stock market's relationship and between oil-importing and oil-exporting countries stock markets themselves is time-varying. Moreover, they notice that the response of stock market returns to oil price changes in oil-importing countries changes is more pronounced than for oil-exporting countries during turmoil periods. They also find that the oil-stock dynamic correlations tend to change as a result of the origin of oil prices shocks stemming from the period of global turmoil or changes in the global business cycle and find that oil prices drive significantly the relationship between oil-importing and oil-exporting countries’ stock markets in both high and low correlations regimes.

I find the findings in the paper interesting.

I have the following comments to the authors to improve their paper

1)      The authors should include a Theory Section, to discuss the related theory on the issue.

2)      Methodology is too briefly. They should discuss the methodology more.

3)      Table 1 should be in Section 4.

4)      There are some structure problems and mistakes in the text. They should polish their writing carefully.

5)      The format of the references is not consistent. They should check all references carefully.  Also, the references are outdated, they should update the references and include more important references in this area.

6)      The main problem is that the authors only study the correlation relationship among the returns. It is more interesting to find out whether there is any cointegration and linear and nonlinear causality among the markets, e.g. whether there is any cointegration and linear and nonlinear causality between oil-importing and oil-exporting countries stock markets. Thus, I suggest the authors should employ cointegration and linear and nonlinear causality in their analysis.

Author Response

Point 1: The authors should include a Theory Section, to discuss the related theory on the issue.

Response 1: a theory section is introduced in the literature review section

Point 2:  Methodology is too briefly. They should discuss the methodology more.

Response 2: in the methodology section we briefly represent the methodological framework employed in the current research. All others methodologies employed in the same research area are introduced and discussed in the literature review jointly with the main findings of anterior empirical studies. Thus, the literature review introduces a background of methodological approach and discusses for each approach the main advantages and disadvantages.

Point 4:   Table 1 should be in Section 4.

Response 4: Table 1 was integrated just before section 4

Point 5: The format of the references is not consistent. They should check all references carefully.  Also, the references are outdated, they should update the references and include more important references in this area.

Response 5: the format of references has been revised.

Point 6: The main problem is that the authors only study the correlation relationship among the returns. It is more interesting to find out whether there is any cointegration and linear and nonlinear causality among the markets, e.g. whether there is any cointegration and linear and nonlinear causality between oil-importing and oil-exporting countries stock markets. Thus, I suggest the authors should employ cointegration and linear and nonlinear causality in their analysis.

Response 6: We find that this proposal is very important, especially when we investigate the relationship between stock markets and oil-stock markets nexus. For this, we thank the reviewer for this precious proposal. However, we think that this proposition could not be added in this paper and it could be considered in future separate paper, admitting that this proposition may lead to a great p-value in the empirical debates about the oil-stock markets nexus.

Round 2

Reviewer 1 Report

Good efforts have been made to address comments.

Author Response

thanks.

Reviewer 2 Report

I suggest the authors polish their writing carefully.

Author Response

Please find the latest version of the paper, ID  economies-472934 with the English editing certification (in attachment) after English editing at MDPI services. 
